# Semantic Space Grounded Weighted Decoding for Multi-Attribute Controllable Dialogue Generation

**Zhiling Zhang**     **Mengyue Wu**[*]
Shanghai Jiao Tong University
Shanghai, China
{blmoistawinde,mengyuewu}@sjtu.edu.cn

**Kenny Q. Zhu**[*]
University of Texas at Arlington
Arlington, Texas, USA
kenny.zhu@uta.edu

## Abstract

Controlling chatbot utterance generation with multiple attributes such as personalities, emotions and dialogue acts is a practically useful but under-studied problem. We propose a novel framework called DASC that possesses strong controllability with a weighted decoding paradigm, while improving generation quality with the grounding in an attribute semantics space. Generation with multiple attributes is then intuitively implemented with an interpolation of multiple attribute embeddings, which results in substantial reduction in the model sizes. Experiments show that DASC can achieve high control accuracy in generation task with the simultaneous control of 3 aspects while also producing interesting and reasonably sensible responses, even in an out-of-distribution robustness test. [1]

## 1 Introduction

Personalized dialogue systems are promising NLP applications for human-computer interaction and emotional companionship. We would expect such systems to have personalities, exhibit emotions, take dialogue acts and even adopt sophisticated strategies (Liu et al., 2021b), which necessitates the research efforts on *Controllable Text Generation*. Despite recent progress in this field (Dathathri et al., 2020; Keskar et al., 2019; Krause et al., 2021), they mainly tackle single-attribute control, overlooking the fact that human interaction can usually convey multiple attributes simultaneously. Therefore, we explore a novel task of *Multi-Attribute Controllable Dialogue Generation*, which can significantly ameliorate the expressiveness, human-likeness, and explainability of chat-bots. However, the numerous combinations of attributes can make the available data for each setting scarce, which poses a great challenge for this task.

Among previous works, *Weighted Decoding* methods has achieved great success in single-attribute control tasks (Arora et al., 2022; Liu et al., 2022). Weighted decoding methods learn a token-level attribute classifier, which predicts the probability of the text conveying the desired attribute given the generation of each token in the vocabulary. Then the predicted probabilities are used to re-weigh the token generation during decoding to induce the attribute. Despite success in single-attribute control, they have certain limitations when extended to the multi-attribute case by multiplying several attribute predictions from multiple classifiers. Extra parameters proportional to the large vocabulary size $|V|$ will be introduced, which can grow several times further due to the number of attributes. The consequent large number of parameters will not only make the model inefficient, but also harm the generation quality. The model can be prone to overfit since the data for each attribute combination are usually small, which increases the risk of degeneration (Holtzman et al., 2020).

To overcome these limitations, we propose **D**ialog **A**ttribute **S**pace **C**ontroller (**DASC**). We establish an attribute semantic space where each token in the vocabulary is projected to the space through *Attribute Token Embedding* shared across attributes. The language models' hidden states are also converted to *Attribute Context Embedding* in the space through attribute-specific layers. The attribute space will be trained to make the tokens suitable to convey the desired attribute close to the current context embedding. We can then assign higher weights for the those tokens during decoding. We will show that DASC can inherit the strong controllability of weighted decoding, while also achieving a natural solution of multi-attribute control with the interpolation of multiple attribute embeddings in the space. Moreover, the shared attribute token embedding also alleviates over-parameterization, and improves the robustness of the model.

---

[*]Corresponding Authors.

[1]Code and data are available at https://github.com/blmoistawinde/DASC.

**Context (Last Rounds):**

A: I often don't get to spend time with my family, and I feel guilty.
B: It's okay. They'll understand you.

**Desired Attributes:**

Gender: Male | Emotion: Fear | Non-Question

**Responses:**

Baseline: Yeah, I will.

Prev Method: I am afraid that my wife and children and my children would be seen by my parents.

DASC: But I am afraid that my wife and children will worry about me.

Figure 1: An example of multi-attribute controllable dialogue generation. The baseline system doesn't attempt any control and produced a dull response, while a previous method of attribute control generated a repetitive and illogical text. DASC successfully gives a response that is both fluent and correctly attributed.

We experiment on an attribute-rich open-domain dialogue dataset (Xu et al., 2022) for the simultaneous control of 3 attribute aspects: Gender Style (male, female, neutral), Emotion (8 classes), and a simple division of Dialogue Act (question VS non-question). As exemplified in Figure 1, compared to previous methods, DASC achieves strong controllability while avoiding low-quality generations in the compositional controlling task. Visualization of the attribute token embeddings (in Figure 3) exhibits specific patterns that benefit the controlling, compared to the general LM token embeddings. We further conducted a robustness test in a out-of-distribution setting and validated that DASC's controllability generalizes. Our contributions are as follows: 1) We propose semantic space grounded weighted decoding for controllable dialogue generation, which can intuitively solve the multi-attribute control task with the interpolation of embeddings in the space; 2) DASC uses smaller number of parameters than other weighted decoding alternatives while achieving better performance with the design of shared attribute embeddings; 3) DASC can achieve high accuracy on the simultaneous control of 3 aspects while also preserving competitive generation quality in both conventional test settings and out-of-distribution robustness tests.

## 2 Method

In this section, we will first define our task and weighted decoding method for controllable gener-

ation as background. Then we will introduce the proposed **DASC** framework.

### 2.1 Task Definition

Given a dialogue **context** $C$ and **attributes** $A = (a_1, a_2, ..., a_K)$, *Controllable Dialogue Generation* aims to generate a **response** $R = (r_1, r_2, ..., r_N)$ that is consistent with the context and carries the attributes.[2] There can be multiple **aspects** grouping the attributes, where in this work we will focus on *Gender Style, Emotion, and Dialogue Act*. An aspect covers multiple related attributes, such as *happiness, sadness* for the Emotion aspect. Each attribute can take three values: 1 means to use the attribute, 0 means not to use and $\phi$ means the attribute is not applicable to the response.

### 2.2 Weighted Decoding for Controllable Generation

Standard, non-controllable, dialogue generation can be formulated with the standard conditional language modeling objective: $L_{CLM} = -\sum_{n=1}^{N} log P(r_n|r_{1:n-1}, C)$

We can use a transformer-based encoder-decoder architecture like BART (Lewis et al., 2020) to model this, where the encoder encodes the context into hidden states as condition for the decoder to generate the response. We will omit $C$ below for brevity.

In controllable dialogue generation, we additionally introduce attributes in the generation condition. Suppose we are generating with a single attribute $a$, then the objective is to model $P(r_n|r_{1:n-1}, a)$. Using Bayes' rule, this can be converted to:

$$P(r_n|r_{1:n-1}, a) \propto P(r_n|r_{1:n-1})P(a|r_{1:n-1}, r_n)^{\alpha} \quad (1)$$

where $\alpha$ is a hyperparameter that can adjust the *control strength*. This means that we can decompose the generation probability into the standard CLM probability weighted by the prediction of another token-wise attribute classifier during decoding. Methods established on such decomposition are thus called **Weighted Decoding** models.

Director (Arora et al., 2022), a representative weighted decoding method, implements the attribute classifier as a linear layer on top of the decoder hidden states. A binary classification is performed on determining whether the generated sentence reflects the desired attribute (e.g. happy or

---

[2]In our work, we make a pre-assumption that attributes are provided by a dialogue policy, and do not include end-to-end scenarios.

not) at each step. For tokens in the sentence from training set, they can be trained with the attribute of the whole sentence using Binary Cross Entropy (BCE) loss. We denote this token-level loss as $L_t$.

$$
\begin{aligned}
L_t &= BCE(P(a|r_{1:n-1}, r_n)) \\
&= BCE(\sigma([W_a h_n]_{r_n}))
\end{aligned} \quad (2)
$$

where $h_n \in \mathbb{R}^d$ is the hidden state for the $n$-th token, $W_a \in \mathbb{R}^{|V| \times d}$ is the learnable weight matrix for attribute prediction given the generation of each token in the vocabulary, and $[\cdot]_{r_n}$ denotes the index selection with the next token $r_n$. Note that it only gathers the attribute logits with the token $r_n$ in the ground truth response. For the other $|V| - 1$ tokens in the vocabulary $V$, they have no label and cannot get trained. Therefore, it uses an extra regularizer to train the prediction on these tokens to as close to 0.5 as possible with MSE loss.

When dealing with multi-attribute control, we can extend Eq. (1) by introducing the product of multiple attribute classifiers, assuming the conditional independence of attributes:

$$
P(r_n|r_{1:n-1}, a) \propto P(r_n|r_{1:n-1}) \prod_{\substack{k=1 \\ a_k \neq \phi}}^{K} P(a_k|r_{1:n})^{\alpha} \quad (3)
$$

The product of probabilities is usually implemented with the summation of logits:

$$
\delta(r_n|r_{1:n-1}, a) = \delta(r_n|r_{1:n-1}) + \alpha \sum_{\substack{k=1 \\ a_k \neq \phi}}^{K} \delta(a_k|r_{1:n}) \quad (4)
$$

Existing works have implemented such an extension with either multiple forward passes through an attribute-conditioned language model (Lin and Riedl, 2021) or one pass of multiple models (Liu et al., 2021a), which can all be very costly as the number of attributes grows. Here we introduce a relatively simple extension of Director, where we just add $K$ linear classifier heads to make the prediction of multiple attributes. We will refer to this simple extension as M-Director, or just Director for simplicity. Note that though more efficient than previous methods, M-Director will still introduce $d \times |V| \times K$ extra parameters. Given that $|V|$ is usually as large as tens of thousands, this model will have enormous number of parameters making it inefficient to train or infer, and also prone to overfitting.

## 2.3 Dialogue Attribute Space Controller

We hypothesize that the above typical methods of weighted decoding may not be the most efficient approach to learn the token-level attribute semantics, especially in multi-attribute cases. The learning objective is imposed on a single token in the target sentence, while all other tokens are regularized equally. This is not usually reasonable, as some tokens similar to the target token should also have high probabilities given the attribute while other tokens different from it are less likely to be generated. For example, for the first token in a *happy* response "nice to meet you", "glad" will also be a reasonable alternative, while "sad" is not, but their attribute label in the training will both be 0.5.

We can fix this counter-intuition in a high-dimensional space. On the one hand, each token has an $p$-dim embedding that encodes its attribute semantics (*Attribute Token Embedding, $ATEMB$*). On the other hand, the hidden states from the LM ($h_n$) are also projected to the same space with attribute-specific linear layers ($W^k \in \mathbb{R}^{p \times d}$) to get *Attribute Context Embedding*, $\hat{h}_n^k = \hat{W}^k h_n$. Thus different vectors in the space convey different semantics, and we call this space *Attribute Semantic Space*.

To leverage this latent space for weighted decoding, for each $\hat{h}_n^k$, we find its attribute-related tokens according to embedding similarity in the space, and assign them higher weights during decoding. Specifically, it is accomplished with a dot-product based token-level attribute classifier.

$$
\delta(a_k|r_{1:n}) = \hat{h}_n^k \cdot ATEMB(r_n) \quad (5)
$$

In this case, when a token is trained with high probability for certain attribute, its neighbors in the attribute space will also have higher probabilities. This alleviates the limitation of previous weighted decoding methods, and eliminates the need for regularizers on other tokens. Further, when applying this to multi-attribute weighted decoding, we get:

$$
\begin{aligned}
\delta(r_n|r_{1:n-1}, a) = \delta(r_n|r_{1:n-1}) \\
+ \alpha(\frac{1}{K} \sum_{\substack{k=1 \\ a_k \neq \phi}}^{K} \hat{h}_n^k) \cdot ATEMB(r_n)
\end{aligned} \quad (6)
$$

where the parenthesized part in the second term can be interpreted as the average/equal-weight interpolation of multiple attribute context embeddings. [3]

---

[3]It is possible to assign different weights for each embedding in interpolation, and we leave it for future works.

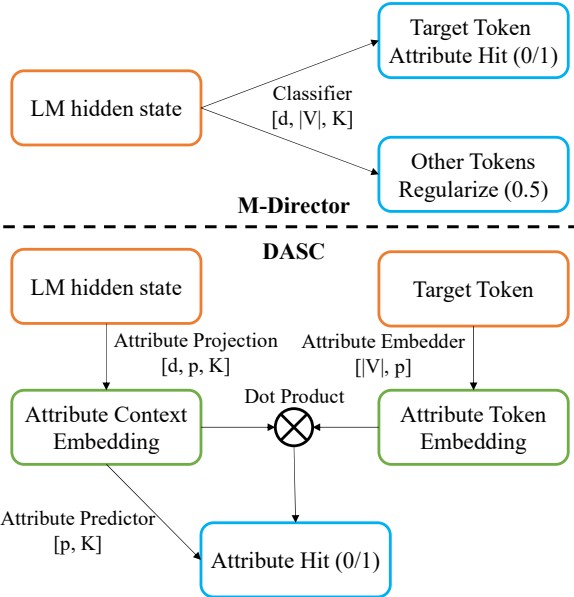

Figure 2: Framework comparison between M-Director and DASC. M-Director uses a classifier head to conduct binary attribute hit classification for each token in the target sentence, and impose regularization for other tokens. DASC projects both LM hidden state and the target token to the attribute space, and uses their dot product for the classification of attribute hit. For each parameterized model component, we show its shape in square brackets.

This formulation suggests that if the attribute space is properly learned and represented, the embedding interpolation will precisely reflect the semantics of the desired attributes, and then DASC can realize reasonable attribute combinations.

To assist the learning of attribute embeddings, we introduce another linear layer on top of the attribute context embedding at each step to directly predict the attributes of the complete response. This can help better align the attribute context embeddings with the corresponding region for its attributes. We denote the new the sentence-level classification loss as $L_s$. For clarity, we give its formulation in the single-attribute case, which can be simply extended to multi-attribute scenarios with the summation over all non-empty attributes.

$$
\begin{aligned}
L_s &= BCE(P(a|r_{1:n-1})) \\
&= BCE(\sigma(v_a \cdot \hat{h}_n))
\end{aligned}
\tag{7}
$$

where $v_a \in \mathbb{R}^p$ is the learnable weight for attribute prediction. Compared with $L_t$ (Eq. (2)), it is a sentence-level classification task independent of $r_n$, which can also be interpreted as predicting the prior probability of the attribute before deciding the

next token to generate, and thus the parameters do not scale with $|V|$. Then the final loss is: $L_{train} = L_{CLM} + \beta(L_s + L_t)$, where $\beta$ is a hyperparameter that controls the weight of attribute-related losses.

We name the proposed framework as Dialogue Attribute Space Controller (**DASC**). The illustration of DASC and its comparison with M-Director is shown in Figure 2. DASC introduce fewer parameters ($d \times p \times K + |V| \times p$) than M-Director ($d \times |V| \times K$). Since we set $p << |V|$, the parameters of attributes projections will be much smaller. And when we deal with $K > 1$, the shared token embeddings across attributes will also save parameters, while the parameters of attribute predictor are almost negligible.

## 3 Experiments

In this section, we will conduct experiments to examine the following hypotheses: (1) DASC can achieve strong controllability while also preserving good generation quality in multi-attribute controllable generation; (2) DASC's performance benefits from the meaningful representations in the attribute semantic space, and reduction in parameters; (3) DASC can also be flexibly extended for other control tasks like the composition of multiple emotions or adopting certain strategies for emotional support.

### 3.1 Experiment Settings

We conduct experiments on the *self* split of the DuLemon dataset (Xu et al., 2022), which is a Chinese open-domain dialogue dataset that is rich in personalized content so that we can find the various attributes we would like to control. We split the data to train/dev/test set into 352,999, 2439, 2412 utterances each. Since the original dataset do not contain annotations of control attributes, we develop a few classifiers, one for each type of attributes, to label the dataset. For gender style (male, female, neutral), we use the dataset released by Su et al. (2020) to train a MacBERT classifier (Cui et al., 2020), which achieved accuracy=94.98%. For emotion, we follow Zhou et al. (2018) and use the NLPCC2013 and NLPCC2014 dataset (8 emotion classes) to train another MacBERT classifier, which has an accuracy of 93.96%. For the question dialogue act (question VS non-question), we simply use a heuristic for labeling: if the sentence contains a question mark(?) we will consider it a question and otherwise non-question. We then use these 3 classifiers to assign each response in the

dataset with the 3 types of attributes (13 of them in total).

### 3.1.1 Competing Methods

We compare the proposed DASC framework with representative methods from different types of controllable generation methods. We use the `fnlp/bart-base-chinese` (Shao et al., 2021) model as the backbone for all competing methods [4]: **Baseline** Simply fine-tuning the backbone on the dataset without utilizing the control attributes. **Rerank** Using top-$p$ sampling (Holtzman et al., 2020) on the baseline model to produce 5 response candidates for each context, and attribute classifiers (here are the same separate models we've used for auto-annotations) to rerank the candidates. Following Thoppilan et al. (2022), we use the sum of predicted probabilities in each aspect for ranking. **CTRL** We re-implemented Keskar et al. (2019)'s method for dialogue generation by defining 3 groups of special control codes for each aspect, and appending the corresponding 3 attribute tokens to each dialogue context during fine-tuning. **Director** The multi-attribute extension of Director (Arora et al., 2022) discussed in Sec. 2.2. We provide more experimental details in Appendix.

### 3.1.2 Evaluation

**Automatic Evaluation** To evaluate the controllability, we use the same attribute classifiers as those used for labeling the dataset to calculate the accuracy of attributes in the generation ($Acc_G$, $Acc_E$, $Acc_Q$ for gender, emotion and question, respectively). For the generation quality, we use BertScore (BScore) (Zhang et al., 2020) to evaluate generation's similarity to reference response, and Distinct-2 (Li et al., 2016) for diversity.

**Human Judgement** We sampled 100 contexts from the test set for human evaluation. Since the distribution of the original test set is extremely skewed, we've specified a constraint for more balanced distribution over all emotions during sampling, so as to ensure the representativeness of the evaluation (21 none, 16 sadness, 16 disgust, 16 happiness, 16 like, 5 anger, 5 surprise, 5 fear). We

---

[4] We also conduct experiments with those leveraging persona description texts, including BoB (Song et al., 2021) and prompting with ChatGPT (OpenAI, 2022), although they may not be especially suitable for controlling the sparse attributes here. We will provide the experimental details and automatic evaluations in Appendix.

|  | BScore | Dist-2 | $Acc_G$ | $Acc_E$ | $Acc_Q$ |
|---|---|---|---|---|---|
| Baseline | 68.18 | 19.25 | 68.49 | 46.31 | 69.61 |
| Rerank | 69.23 | 19.28 | 75.46 | 54.93 | 82.42 |
| CTRL | **71.09** | 18.91 | 85.32 | 77.49 | **100.00** |
| Director | 69.54 | _21.40_ | _95.81_ | **86.73** | **100.00** |
| DASC | _70.42_ | **21.94** | **95.85** | _86.07_ | **100.00** |

Table 1: Automatic evaluation results on DuLemon test set. The best results are in bold, while the second results are underlined.

invited 2 volunteers who are native Chinese speakers to evaluate each generation from 3 perspectives. **Attribute Accuracy**: if the response conveys the given attribute. **Sensibleness**$_{(1-4)}$: if the response is fluent, coherent with the context, and accords with commonsense. **Interestingness**$_{(1-4)}$: whether the response is specific, novel and can encourage more interesting conversation continuation.

### 3.2 Results

Automatic evaluation results are shown in Table 1. We can see that *Rerank* failed to show strong controllability because the base model struggles to produce attributed ranking candidates without finetuning with the attributes. *CTRL* leveraged the attributes in finetuning, and achieved better control accuracy and BertScore, but it doesn't produce more diverse responses overall. Both *Director* and *DASC* exhibit the best controllability, and *DASC* produces more diverse and reasonable responses according to Distinct-2 and BertScore.

|  | $Acc_G$ | $Acc_E$ | $Acc_Q$ | Interest | Sensible |
|---|---|---|---|---|---|
| Baseline | 0.80 | 0.55 | 0.64 | 2.04 | **3.46** |
| Rerank | 0.81 | 0.62 | 0.82 | 2.13 | 3.44 |
| CTRL | 0.85 | 0.82 | **0.97** | 2.24 | **3.46** |
| Director | 0.87 | 0.87 | 0.96 | 2.25 | 3.26 |
| DASC | **0.88** | **0.88** | **0.97** | **2.37** | 3.28 |

Table 2: Human Judgement on DuLemon test set.

We then show human judgement results in Table 2. The inter-annotator agreement for $Acc_G$, $Acc_E$ and $Acc_Q$ are 0.65, 0.55 and 0.64 in Cohen's $\kappa$, which indicates moderate to substantial agreement. The agreement of $Interestingness$ and $Sensibleness$ is 0.48 and 0.44 in Pearson's $r$. This is hardly surprising because the latter two perspectives are highly subjective. The evaluation on attribute accuracies is similar to the automatic results, except that the accuracy of gender drops slightly. We find that human evaluators can spot errors related to gender stereotype (Bolukbasi et al., 2016), like generating soldier for male style and baby-carer for female, where these occupations should

be gender-neutral. The annotators also check questions without a question mark, which explains the slight difference in $Acc_Q$.

Overall, the rankings of controllability still hold according to human evaluation, with DASC performing the best. Baseline, Rerank and CTRL have slightly better *Sensibleness* than weighted decoding methods, which agrees with the commonly observed controllability-quality trade-off in previous literature (Dathathri et al., 2020; Yang and Klein, 2021; Qian et al., 2022). All controllable generation methods achieved higher *Interestingness* score than baseline, which supports the benefits of controllable generation. DASC achieved the best *Interestingness* given similar attributes accuracy as Director, indicating the effectiveness of attribute semantic space, which can establish better representations of attribute semantics and a more reasonable approach to compose the control attributes in weighted decoding.

### 3.3 Robustness Test

In previous experiments, the control attributes provided to the model come from the reference response. Therefore, models may coincidentally hit the desired attributes when generating the most likely response to the context, without truly reliable controllability for arbitrary given attributes. Hence, we further conduct experiments to test the robustness of the controllable generation methods in out-of-distribution scenarios.

Specifically, we sampled 100 contexts from the test set, and give the models each of the 8 emotions as the generation condition, paired with the original gender and question act[5]. We then use greedy decoding to generate response for each (context, attributes) pair and conduct similar automatic and human evaluation on the 800 generations.

| | Dist-2 | $Acc_E$ | Interest | Sensible |
|---|---|---|---|---|
| Rerank | 17.55 | 17.00 | - | - |
| CTRL | 21.07 | 43.38 | 1.91 | **3.00** |
| Director | *34.73* | 61.88 | 1.62 | 2.27 |
| DASC | **26.71** | **65.38** | **2.08** | 2.82 |

Table 3: Robustness test results.

Table 3 shows the robustness test results.[6] Compared with Table 1, we can see that the emotion ac-

---

[5]We do not change these 2 attributes as they are sometimes determined given the context.

[6]BertScore is not reported here, as the model can be directed towards attributes different from the ground truth, invalidating the similarity-based metric as a proxy for generation quality.

curacy of Rerank and CTRL dropped significantly, which shows that their controllability is not generalizable. Another notable phenomenon is the abnormal *Distinct-2* achieved by Director. We then further analyze their performance with human evaluation (excluding Rerank as it fails to control attributes). We found that Director frequently generate ungrammatical, illogical and repetitive long responses (like the second response in Figure 1). Director's loss in emotion accuracy is also higher than DASC, indicating that it may overfit the training distribution given its large parameters, and thus performs worse in this out-of-distribution setting. Compared to CTRL, DASC has lower *Sensibleness* but higher *Interestingness*, when it also has a significant advantage in diversity and controllability.

### 3.4 Semantic Space Visualization

For a clear understanding of how the proposed attribute semantic space can help controllable generation, we visualize them in 2D space with t-SNE (Van der Maaten and Hinton, 2008). First, we visualize the attribute token embeddings of some representative attribute-related tokens, and also compare them with the corresponding embedding in the original LM (Figure 3). Comparing the two figures, we can see that (1) The token embeddings from different aspects are more separable in the attribute space (see points with different colors), while tokens in the same aspect are closer despite the difference in other linguist features like part-of-speech (like 'handsome' and 'male'). (2) The token embeddings from different attributes of the same aspect are also distinguished in the attribute space (like 'male'-'female', 'love'-'miserable'). These characteristics enable DASC to successfully control the generation of distinctive attributes and compose attributes from different aspects.

Next, we also visualize the attribute context embedding. Specifically, we take the responses with certain attribute in the dev set of the dataset, feed them into the model and average attribute context embeddings at each decoder token as sentence-level representations, and pair them with the sentence-level attribute annotations for analysis. For brevity, we only show the visualization with emotion labels in Figure 4, and provide those with gender and question act labels in Appendix. We can see that the context embeddings from sentences with different emotions are clearly separated in the space, which supports the strong controllability of

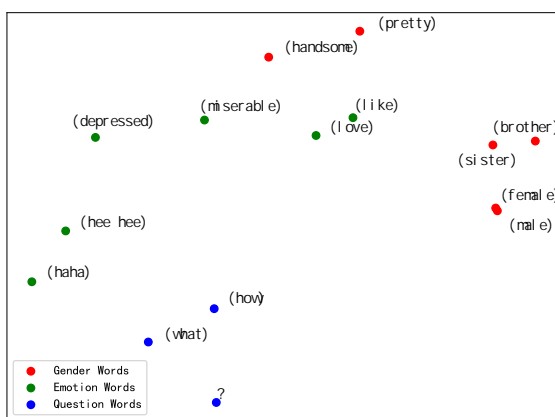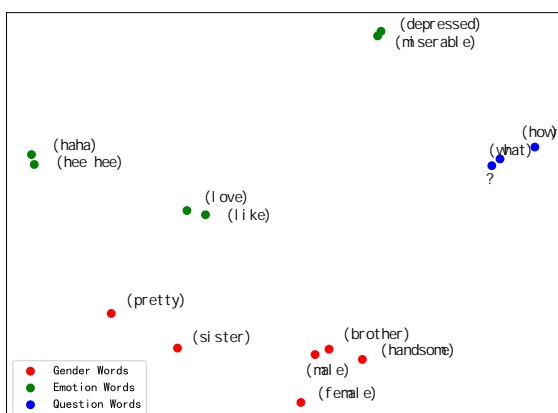

Figure 3: Comparison of two sets of token embeddings with t-SNE visualization: those from the language model (left) and from the attribute semantic space (right).

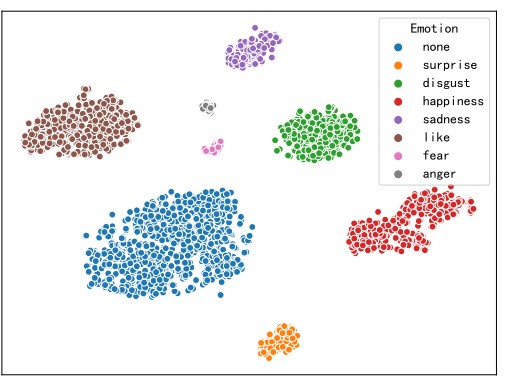

Figure 4: The t-SNE visualization of attribute context embedding of responses with different emotions.

| Method | #params | BScore | Avg Acc |
|---|---|---|---|
| baseline | - | 68.18 | 61.47 |
| DASC ($p$=512) | 15.94M | 70.18 | 92.56 |
| DASC ($p$=1024) | 31.88M | 70.12 | 92.72 |
| DASC ($p$=2048) | 63.75M | **70.42** | 93.97 |
| DASC ($p$=4096) | 127.50M | 70.26 | **94.42** |
| Director | 210.98M | 69.54 | 94.14 |

Table 4: Effect of the number of extra parameters for controllability and generation quality.

model (210.98M vs. 116.26M), may have been over-parameterized and thus harms its generation quality. A moderate size of DASC can achieve the best BertScore, but smaller ones do not significantly degrade the performance. This suggests tha DASC can be a promising candidate in application, given its parameters are fewer than alternatives and are orders of magnitude fewer than LLMs that has generally over 6B parameters.

### 3.6 Case Study

Besides multi-aspect control as shown in Figure 1, we also show a proof-of-concept application that DASC can naturally blend two emotions in one generated response. We can simply achieve this by setting both attributes' value as 1 instead of $\phi$. The results are shown in Figure 5 and Figure 8. We can see that DASC can successfully generate responses with either single emotion or the combination of both emotions, where the later can produce potentially more vivid response.

### 3.7 ESConv Experiment

To further explore the potential of DASC, we also experimented on another dataset ESConv (Liu et al., 2021b). It is an English dataset that aims to pro-

DASC with multiple attributes.

### 3.5 Parameter Analysis

As analyzed before, DASC can use a relatively smaller amount of parameters to implement weighted decoding for multi-attribute controllable generation. Here we study the effect of number of parameters by adjusting the dimension of the attribute space $p$, and comparing with baseline and M-Director which uses no/large amount of parameters for attribute control. We use BertScore to evaluate the generation quality and the average control accuracy on 3 aspects to reflect controllability.

Results are shown in Table 4. Comparing DASC with different $p$, we can see that larger amount of parameters can generally improve the model's controllability, but even a relatively small $p$ ($p$=512) is already capable to achieve high control accuracy. For generation quality, Director, which additionally uses nearly twice the parameters of the base

**Context (Last Rounds):**

> A: Hello, nice to meet you. Aren't you at work?
> B: I'm an ordinary actor. I don't have to shoot these days.

**Desired Attributes:**

Gender: Neutral | Non-Question

**DASC with different emotions:**

> Surprise: Wow. I thought you were a big star.

> Like: That's admirable! I'm still at school.

> Surprise+Like: Wow, that's awesome! I used to be an actor too, but now I'm just an ordinary white-collar worker.

Figure 5: DASC generates different responses to the same context given different emotions and their composition as control attributes.

vide emotional supports to help seekers with 8 defined strategies. Here we use the human annotated strategy labels as the control attributes, and experimented with 3 methods: **Baseline**, **CTRL** and **DASC**. We excluded Director here for its inefficiency. We report the automatic metric **Distinct-2** and human evaluated **Strategy Accuracy**, **Usefulness**$_{(1-4)}$ and **Sensibleness**$_{(1-4)}$. In Table 5, we can see that the control of relatively complex strategies is harder, and thus the accuracy is lower than the previous experiment (Table 2). Nevertheless, DASC still achieves reasonable control accuracy and outperforms other methods on all metrics. These results suggest that DASC is language-agnostic and can be effectively applied to many kinds of attribute controls. We provide more details and generation examples in Appendix.

|          | Dist-2 | Acc  | Useful | Sensible |
|----------|--------|------|--------|----------|
| Baseline | 19.28  | 0.27 | 1.92   | 3.30     |
| CTRL     | 21.20  | 0.52 | 2.04   | 3.31     |
| DASC     | **25.86** | **0.70** | **2.24** | **3.48** |

Table 5: Test results on ESConv.

## 4 Related Work

Controllable generation has gained wide research interest recently. PPLM (Dathathri et al., 2020) proposed a plug-and-play framework to control the generation with an extra attribute classifier. Later research progress can be roughly divided into 3 categories. *Reranking* methods leverage attribute classifiers to either simply rank the full generation candidates (Thoppilan et al., 2022), or partial generations for the guidance of future outputs (Yang and Klein, 2021). *Integrated* methods inte-

grate attribute-related trainable parameters into the generation model for fine-tuning, such as discrete control codes (Keskar et al., 2019) or continuous prompt prefix (Qian et al., 2022). *Weighted Decoding* methods leverage token-level attribute classifiers to guide each decoding step. For example, Krause et al. (2021) and Liu et al. (2021a) utilized one/two additional class conditional language models to provide the attribute discrimination. Director (Arora et al., 2022) integrates the attribute classifier as simple linear layers on top of LM hidden states.

Multi-attribute controllable generation is relatively under-explored now. Lin and Riedl (2021) proposed to extend weighted decoding for the multi-attribute case with the simple product of multiple attribute conditional language models. Gu et al. (2022) proposed a VAE-based method combined with an intersection-searching algorithm for multi-aspect controllable generation, but their method cannot simply apply to conditional generation tasks like dialogue generation. Mireshghallah et al. (2022) proposed an energy-based controllable generation method that can combines multiple controls, but are mainly suitable for fixed-length generation.

Controllable generation techniques are especially important in dialogue systems and the applications of several controlling aspects have been studied. For example, we may condition the generation with dialogue acts for the genuine reflection of the desired behavior (Wen et al., 2015), add emotions in the response to enhance the expressiveness of the bot (Zhou et al., 2018), and also impose personal profiles like gender (Su et al., 2020) and persona (Zhang et al., 2018) to establish a human-like companion. Recent advance in LLMs has pushed the frontiers of dialog generation, enabling applications like role-playing with complex personality and memory (Park et al., 2023). However, their exorbitant cost and privacy concerns make them less relevant in certain deployment scenarios.

Another line of controllable generation utilizes dense persona descriptions (Zhang et al., 2018). This paradigm is capable of expressing rich persona information in free text, such as personal status, hobbies and occupations. The natural language form allows for integration of other language resources for an enhanced generation quality. For example, Song et al. (2021) disentangles the task of persona consistency learning and response generation, and leverages non-dialogue NLI datasets

to help the former and consequently enhance the latter. However, although attributes can also be expressed in free-text descriptions, they can contain noise, and making them less effective than attribute-specific methods, as shown in our experiments (Appendix C). It would be promising to further combine the two paradigms for more general controllable generation (Tang et al., 2023).

## 5 Conclusion

In this paper, we propose DASC, a novel framework for multi-attribute controllable dialogue generation. It is established on the weighted decoding paradigm for strong controllability and further grounds it in an attribute semantic space, which enables the simultaneous control of multiple attributes with the interpolation of multiple attribute embeddings. Experiments show that DASC can achieve strong controllability for multi-attribute generation while also preserving high quality in out-of-distribution scenarios. DASC is highly efficient given its much fewer number of parameters than alternatives and LLMs, making it an promising choice for deployment.

## Limitations

Some limitations of the proposed methods remain to be addressed in future research.

First, our experiment settings assume that the desired attributes are available for generation, which would require a separate dialogue policy to decide the attribute label provided to the model. Therefore, our model cannot be directly applied to end-to-end dialogue models, and may also be affected by the potential error propagation from the dialogue policy model. Since the intended use of DASC is to serve as a component of pipeline-style dialogue systems, these common issues in such systems are out of the scope of this work.

Moreover, we require annotated datasets with multiple attributes for evaluation, which are rare. Therefore, we evaluate the capability of multi-attribute control mostly on one dataset. Experiments on more datasets, especially those with additional attributes may be required to further validated the efficacy of the proposed methods.

Last but not least, DASC is not directly applicable for controllable generation with free text as control signal, such as persona descriptions (Zhang et al., 2018), which might limit its application range, though we may simply combine DASC with other techniques like concatenating the descriptions to achieve this goal, which will require further explorations.

## Ethics Statement

The proposed method is utilized for the control of gender style. As we've noticed and discussed in Sec. 3.2, the model may resort to gender stereotypes for generating responses in that gender. The potential reason is that the dataset used to train the classifier already contains gender-biased labels, and such biases are exploited by the classifier, and passed to the generation model through the automatic annotated labels. To avoid such effects, we may carefully clean the dataset for such biased labels (Gehman et al., 2020), or mine such biased tokens and penalize them during weighted decoding. We may also apply RLHF to further mitigate the biases (Ouyang et al., 2022).

Though the proposed method is mainly intended for improving the interestingness of the chatbot, and endowing the model with abilities like emotional support, such method may also be applied for vicious application. For example, they may introduce toxicity as an attribute and encourage the model to generate more toxic responses. Therefore, the application range of such techniques should be carefully restricted.

We adhere to the license of the used datasets.

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

## A  Effect of Control Strength

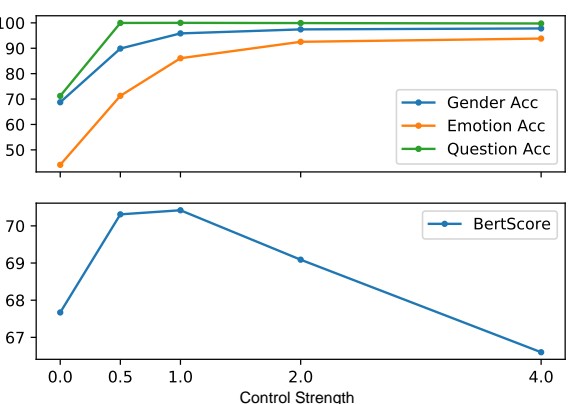

Figure 6: Effect of control strength on controllability and generation quality

We show the effect of control strength $\alpha$ (Eq. (6)) on DASC's controllability and generation quality in Figure 6. From the trend shown in this figure, we can see that for *Question* which is easy to control, we can already achieve perfect control with a low $\alpha$, while harder attributes like *Emotion* would require a higher $\alpha$ to get a high success rate. Therefore, we may further hypothesize that a better balancing of the control accuracy of each attribute and the generation quality can be achieved by setting different control strengths for each aspect, like higher $\alpha$ for Emotion and lower $\alpha$ for Question. Careful tuning of the parameters or specific searching algorithms

(Gu et al., 2022) may serve the goal, and we leave this for future work.[7]

## B Experiment Details

For all experiment methods, they use `bart-base`[8] as the base model. All models are fine-tuned on the dataset for 6 epochs. When conducting multi-aspect control for Director and DASC under weighted-decoding paradigm (Eq. (4)), we set the variables for the desired attribute as 1, and other variables as $\phi$. The decoding method is top-$p$ sampling with $p = 0.5$. DASC uses control weight $\alpha = 1$ and classifier loss weight $\beta = 0.1$, similar as the previous experiments. All experiments of the paper are conducted on a Linux server, and each experiment is run on a single NVIDIA A100 GPU. To avoid overfitting, we select the checkpoint with the best BertScore on dev set for final testing. We fix the random seed in experiment and report the results coming from a single run. Below we provide the specific details for the experiments on ESConv.

For experiments on ESConv (Liu et al., 2021b), we use the latest released version[9], which has 1,300 conversations, and we split them into 1,100/100/100 train/dev/test set, which contains 15,605/1,403/1,369 utterances each. In human evaluation, we further sample 15 utterances for each of the 7 emotional support strategies defined in the dataset (except the vague *Other* class), and get 105 utterances in total. The meaning of **Sensibleness**$_{(1-4)}$ is similar to the experiment in the previous dataset: if the response is fluent, coherent with the context, and accords with commonsense. By **Usefulness**$_{(1-4)}$, we consider if the response dives deep in the problem faced by the support seeker, is comforting, contains useful suggestions or encourages in-depth further discussions.

## C Experiments with Description Control

Dense persona descriptions are another common form of control signal in dialog generation (Zhang et al., 2018), and we can also convert the sparse attributes into descriptive texts to be compatible with these methods. Therefore, we now supplement new experiment results with two representative methods that leverages persona descriptions for control.

---

[7]In practice, we further multiply $\alpha$ by the number of attributes $K$ to adapt to the variable attribute numbers, which is not counted in Eq. (6) and Figure 6 for clarity.

[8]https://huggingface.co/facebook/bart-base

[9]https://github.com/thu-coai/Emotional-Support-Conversation

**BoB** (Song et al., 2021) disentangles the task of persona consistency learning and response generation, and leverages non-dialogue NLI datasets to help boost the performance of consistency learning and finally improves personalized generation.

To apply BoB on the dataset we've experimented with, we convert the discrete attribute annotations into textual descriptions with rules. For example, the male/female gender will be converted to "I'm a girl/boy.", a question will have the description "I want to ask a question". And for emotion, we fill them in the template "I feel {emotion}." We concatenate these 3 description texts as the persona text to be used by BoB. As is suggested in the official GitHub repository, we leverage the Chinese NLI dataset CMNLI (Xu et al., 2020) as the auxiliary inference datasets.

**ChatGPT** (OpenAI, 2022) is a representative Large-Language-Model (LLM) that can follow human instruction and give responses. Therefore, we can encode the attribute values into the ChatGPT system message to achieve control on them. We use the template:

> A dialog history is given below. Please act as the {current speaker} and respond to the next sentence with a {dialog act} in the voice of {gender and emotion}.

Then we use dialog history as the user message, and let ChatGPT (gpt-3.5-turbo) give a following sentence. We use temperature=0 and stop="\n", that is, greedy search for one line of text similar to the setting of the original dataset.

|  | BScore | Dist-2 | Acc$_G$ | Acc$_E$ | Acc$_Q$ |
|---|---|---|---|---|---|
| Baseline | 68.18 | 19.25 | 68.49 | 46.31 | 69.61 |
| CTRL | **71.09** | 18.91 | 85.32 | 77.49 | **100.00** |
| DASC | 70.42 | 21.94 | **95.85** | **86.07** | **100.00** |
| BoB | 65.47 | 23.44 | 74.59 | 64.76 | 98.51 |
| ChatGPT | 66.21 | **30.98** | 69.49 | 56.88 | 98.22 |

Table 6: Automatic evaluation on the DuLemon test set with persona description-based controlling methods.

|  | Dist-2 | Acc$_E$ |
|---|---|---|
| CTRL | 21.07 | 43.38 |
| DASC | 26.71 | **65.38** |
| BoB | 24.25 | 30.13 |
| ChatGPT | **37.02** | 30.00 |

Table 7: Automatic evaluation on the DuLemon robustness test with persona description-based controlling methods.

Firstly, we can see that ChatGPT can generate highly diverse texts (with dist-2 on test set similar as the human ground truth, 30.98 VS 30.63). However, ChatGPT cannot achieve comparable controllability with finetuned methods like CTRL and DASC, especially on attributes whose manifestation cannot be easily described in the prompt (e.g. Gender and Emotion).

Moreover, we can see that BoB also shows strong controllability compared to baseline and zero-shot ChatGPT. It also has higher generation diversity than other methods except ChatGPT, potentially due to the introduction of the auxiliary inference dataset. However, its controllability is relatively worse than DASC. Its generation quality is also poor, both reflected in the low BertScore and our manual check, where we find many influent cases like "Yes. After all, I'm a lawyer, otherwise it would be hard to find a body.". BoB also shows less sensitivity to the change of control signals, as is shown in the lower dist-2 and emotion accuracy in the robustness test.

To conclude, we believe it is possible for description-based controllable generation methods like ChatGPT and BoB to perform better in the control of discrete attributes, but it would require significant efforts on prompt engineering (e.g. describe in more detail for ChatGPT, or make the description more similar to the auxiliary inference dataset for BoB). However, when we have relatively sufficient labeled data the discrete control attributes, DASC on 'small' LMs will certainly be a simple and competitive choice.

## D Examples and Visualizations

In this section, we provide supplementary examples and figure visualizations.

Figure 7 shows the original Chinese text for the emotion composition example in Figure 5, and we provide another example in Figure 8, which shows that DASC can even compose a positive emotion *Like* and a negative emotion *Fear* in the same response to express complex meanings.

We provide the t-SNE visualizations of attribute context embeddings of sentences with different *Gender Style* and *Question Act* in Figure 9 and Figure 10. We find similar results as we've seen for emotion, that the embeddings from different attributes are clearly separated.

We also show 2 examples of the generated results on the ESConv. In Figure 11, both baseline

**Context (Last Rounds):**

A: 你好，很高兴认识你，你没上班吗？
B: 我是个小演员，最近不用拍戏。

**Desired Attributes:**

Gender: Neutral | Non-Question

**DASC with different emotions:**

Surprise: 原来是这样啊，我还以为你是大明星呢。

Like: 真羡慕你，我还在上学呢。

Surprise+Like: 哇塞，太厉害了，我以前也是个演员，不过现在只是一个小白领。

Figure 7: Original Chinese text for Figure 5

**Context (Last Rounds):**

A: 你喜欢看什么电影呢？ (What movies do you like to watch?)
B: 我比较喜欢看恐怖片，你喜欢吗？ (I prefer to watch horror movies. Do you like them?)

**Desired Attributes:**

Gender: Neutral | Non-Question

**DASC with different emotions:**

Fear: 我不喜欢，我觉得恐怖片太吓人了。 (I don't like them. I think they're too scary.)

Like: 我也喜欢看恐怖片，我觉得看着特别刺激。 (I also like to watch horror movies, I think it is very exciting.)

Fear+Like: 我也挺喜欢的,但是我不敢看。 (I also like them, but I'm too scared to watch them.)

Figure 8: Example of DASC composing *Like* and *Fear* emotion in the generated response.

and DASC successfully applied the desired strategy, while CTRL failed to do so. However, baseline also included a repetitive question at the end, while DASC gives a more comprehensive restatement, which exhibits a deep understanding of the situation and will be regarded as more helpful for the help seeker. In Figure 12, only DASC used the correct strategy in the generation, and such precise reflection of the anxious mood makes the response more sympathetic.

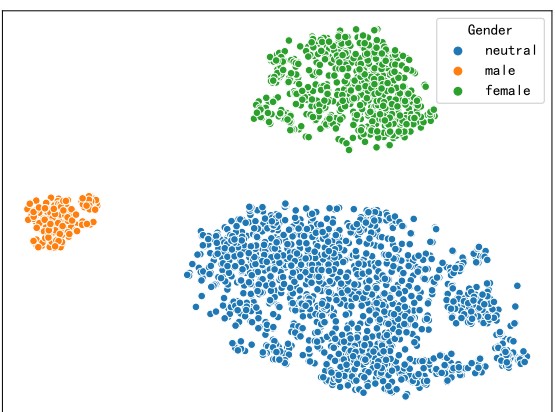

Figure 9: Visualization of attribute context embedding of responses with different gender styles.

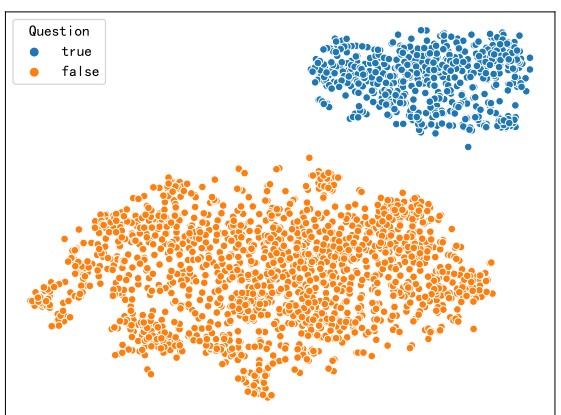

Figure 10: Visualization of attribute context embedding of responses with different question acts.

**Context (Last Rounds):**

A: Glad to know that you are doing well. How may I assist you today ?
B: I had a drinking challenge which has been affecting my marriage for quit sometime now

**Strategy:** [Restatement or Paraphrasing]

**Responses:**

Baseline: I am sorry to hear that you are having a drinking challenge. How may I assist you today?

CTRL: I am sorry to hear that. I have been in a similar situation before.

DASC: I see. So you are having a drinking challenge and you are struggling to get your marriage to work?

Figure 11: System generations in one example of ES-Conv, with the "Restatement or Paraphrasing" strategy.

**Context (Last Rounds):**

A: What's on your mind?
B: I have real bad anxiety about the possible 2nd shut down due to the rise in Covid.
B: So many people lost their job during the first shut down

**Strategy:** [Reflection of feelings]

**Responses:**

Baseline: I'm so sorry to hear that. Have you tried to talk to your boss about it?

CTRL: I'm sorry to hear that. I'm sure you will get through this.

DASC: I can understand how you are feeling, I've been in this position myself and I can tell you are very anxious.

Figure 12: System generations in one example of ES-Conv, with the "Reflection of feelings" strategy.