# OpenReview forum: "Semantic Space Grounded Weighted Decoding for Multi-Attribute Controllable Dialogue Generation"
_EMNLP/2023/Conference — EMNLP 2023 Main_

### Official Review · Reviewer_yeTY · 2023-07-26

**Soundness:** 4

**Excitement:**

4: Strong: This paper deepens the understanding of some phenomenon or lowers the barriers to an existing research direction.

**Missing References:**

Many models do not get a chance to compare, such as BoB: BERT Over BERT for Training Persona-based Dialogue Models from Limited Personalized Data (Song et al., ACL-IJCNLP 2021).

This work Enhances Personalized Dialogue Generation with Contrastive Latent Variables: Combining Sparse and Dense Persona (Tang et al., ACL 2023) is similar to the author's work, except that it addresses "multi-attribute controlled conversation generation" in text descriptions, which the authors can try to focus on. Of course, this is just advice, I wouldn't ask for it (in the three months before the deadline).

**Paper Topic And Main Contributions:**

The authors aim to generate personalized conversations (gender, mood, etc.) through techniques for controlled conversation generation. They proposed DASC(Dialog Attribute Space Controller) to control multiple attributes at the same time to generate personalized replies, that is, multi-attribute controlled conversation generation. The author proposes a new weighted decoding method. Each lexeme in the vocabulary is mapped to an Attribute semantic space through the Attribute Token Embedding. The hidden state of the language model is also transformed into the Attribute Context Embedding in the space through the attribute-specific layer. Then, a dot-product-based attribute classifier assigns higher weights to adjacent words embedded in the current context in the attribute space, thus achieving weighted control in the decoding process. Experiments on open domain dialogue datasets show that DASC achieves strong controllability in combinatorial control tasks, the visualization of attribute lexical embedment clearly shows the specific pattern related to control, and the controllability of DASC for a single attribute is also guaranteed.

**Reasons To Accept:**

Different from previous work, the author deals with the attributes of characters separately and turns it into a multi-attribute controllable dialogue generation task, which enables us to explore how different attributes affect dialogue generation, and the author makes this process very clear, which is very helpful for subsequent work.

**Reasons To Reject:**

The author has little exploration of the previous works. The experiment is not comprehensive enough. Most of the comparison of the author's experimental results is based on the accuracy of attribute prediction. Still, some models do not adopt this method to deal with attributes (such as overall processing) and can obtain good generated results, and these models have not been compared.

**Reproducibility:**

4: Could mostly reproduce the results, but there may be some variation because of sample variance or minor variations in their interpretation of the protocol or method.

**Reviewer Confidence:**

3: Pretty sure, but there's a chance I missed something. Although I have a good feel for this area in general, I did not carefully check the paper's details, e.g., the math, experimental design, or novelty.

---

> ### Author Rebuttal · Authors · 2023-08-28
>
> Thank you for providing a broader view on controllable generation by providing more related works like BoB. We are aware of this excellent work that leverage textual persona description for controllability, but our work focuses on sparse persona attributes, which makes it not easy to establish a fair comparison. It would certainly be interesting to compare the two lines of works. Therefore, we present our trial on comparing BoB with our methods. We will show the results first, and give the experimental details at the bottom.
>
> *Since it is difficult to make fair comparison with prior results using human annotated scores, we just provide the automatic metrics. According to the suggestion of other reviewers, we also added another baseline ChatGPT, where we describe the attributes in prompt for zero-shot generation.
>
> Results on the test set (ref Table 1).
>
> |          |   BScore  |   Dist-2  |   Acc_G   |   Acc_E   |    Acc_Q   |
> |:--------:|:---------:|:---------:|:---------:|:---------:|:----------:|
> | Baseline |   68.18   |   19.25   |   68.49   |   46.31   |    69.61   |
> |   CTRL   | **71.09** |   18.91   |   85.32   |   77.49   | **100.00** |
> |   DASC   |   70.42   |   21.94   | **95.85** | **86.07** | **100.00** |
> |   BoB    |   65.47   |   23.44   |   74.59   |   64.76   |    98.51   |
> |  ChatGPT |   66.21   | **30.98** |   69.49   |   56.88   |    98.22   |
>
> Robustness test results (ref Table 3).
> |         |   Dist-2  |   Acc_E   |
> |--------:|:---------:|:---------:|
> |    CTRL |   21.07   |   43.38   |
> |    DASC |   26.71   | **65.38** |
> |     BoB |   24.25   |   30.13   |
> | ChatGPT | **37.02** |   30.00   |
>
> We can see that BoB also shows strong controllablity compared to baseline and zero-shot ChatGPT. It also has higher generation diversity than other methods except ChatGPT, potentially due to the introduction of the auxiliary inference dataset. However, its controllablity is relatively worse than DASC. Its generation quality is also poor, both reflected in the low BertScore and our manual check, where we find many influent cases like "Yes. After all, I'm a lawyer, otherwise it would be hard to find a body."
>
> BoB also shows less sensitivity to the change of control signals, as is shown in the lower dist-2 and emotion accuracy in the robustness test.
>
> To conclude, we believe it is possible for text-based controllable generation methods like ChatGPT and BoB to perform better in the control of discrete attributes, but it would require significant efforts on prompt engineering. However, when the discrete control attributes are already available, DASC will certainly be a simple and competitive choice.
>
> We will also add more discussions about this line of work like (Tang et al., ACL 2023) in related works.
>
> ---
>
> *Experimental Details*: To apply BoB on the dataset we've experimented with, we convert the discrete attribute annotations into textual descriptions with rules. For example, gender=female will be converted to "I'm a girl.", a question will have the description "I want to ask a question". And for emotion, we fill them in the template "I feel {emotion}." We concatenate these 3 description texts as the persona text to be used by BoB. We leverage the NLI dataset CMNLI as the auxiliary inference datasets.

---

### Official Review · Reviewer_fjTP · 2023-08-05

**Soundness:** 4

**Excitement:**

3: Ambivalent: It has merits (e.g., it reports state-of-the-art results, the idea is nice), but there are key weaknesses (e.g., it describes incremental work), and it can significantly benefit from another round of revision. However, I won't object to accepting it if my co-reviewers champion it.

**Paper Topic And Main Contributions:**

This paper proposes a controllable dialogue generation model. Specifically, it aims to control multiple attributes of the generated dialogue responses. The author introduces a method named Dialog Attribute Space Controller (DASC). They try to directly modify the logits of the generated sequences with some attribute classifiers and some soft labels produced by the similarity between the target attribute and next tokens. Evaluation results show that the proposed method can control the attribute of the generated responses.

**Reasons To Accept:**

1.  An interesing task is investigated.

2. An effective method is proposed. The experiments results show that the proposed method help to enhance the controlability of different attributes.

**Reasons To Reject:**

1. The experiment section could be improved. For example, it is better to carry significance test on the human evaluation results. It is also beneficial to compare the proposed method with some most recent LLM.

2. The classifier of determining attributes using only parts of the sentence may not perform well. Specifically, I am wondering what is the performance of the attribute classifer obtained using Eq.2 and Eq.7.

3. Some of the experiment results could be explained in more details. For example, the author observes that "Compared to CTRL, DASC has lower Sensibleness but higher Interestingness", but why? Is that because DASC is bad for exhibiting Sensibleness? Similar results are also observed in Table1.

**Reproducibility:**

4: Could mostly reproduce the results, but there may be some variation because of sample variance or minor variations in their interpretation of the protocol or method.

**Reviewer Confidence:**

4: Quite sure. I tried to check the important points carefully. It's unlikely, though conceivable, that I missed something that should affect my ratings.

---

> ### Author Rebuttal · Authors · 2023-08-28
>
> Thanks for your constructive reviews.
>
> ### 1. The experiment section could be improved. For example, it is better to carry significance test on the human evaluation results. It is also beneficial to compare the proposed method with some most recent LLM.
>
> We agree that examining the performance of the most recent LLM on this task would be interesting. Therefore, we now supplement new experiment results with ChatGPT(gpt-3.5-turbo) on the DuLemon dataset. We will show the results first, and give the experimental details at the bottom.
>
> *Since it is difficult to make fair comparion with prior results using human annotated scores, we just provide the automatic metrics. According to the suggestion of reviewer yeTY, we also added another baseline BoB, which is a representative method with finetuned model that leveraged persona description text for controllable generation.
>
> Results on the test set (ref Table 1).
> |          |   BScore  |   Dist-2  |   Acc_G   |   Acc_E   |    Acc_Q   |
> |:--------:|:---------:|:---------:|:---------:|:---------:|:----------:|
> | Baseline |   68.18   |   19.25   |   68.49   |   46.31   |    69.61   |
> |   CTRL   | **71.09** |   18.91   |   85.32   |   77.49   | **100.00** |
> |   DASC   |   70.42   |   21.94   | **95.85** | **86.07** | **100.00** |
> |   BoB    |   65.47   |   23.44   |   74.59   |   64.76   |    98.51   |
> |  ChatGPT |   66.21   | **30.98** |   69.49   |   56.88   |    98.22   |
>
> Robustness test results (ref Table 3).
> |         |   Dist-2  |   Acc_E   |
> |--------:|:---------:|:---------:|
> |    CTRL |   21.07   |   43.38   |
> |    DASC |   26.71   | **65.38** |
> |     BoB |   24.25   |   30.13   |
> | ChatGPT | **37.02** |   30.00   |
>
> We can see that ChatGPT can generate highly diverse texts. However, ChatGPT cannot achieve comparable controllability with finetuned methods like CTRL and DASC, especially on attributes whose manifestation cannot be easily described in the prompt (e.g. Gender and Emotion).
>
> Therefore, we may conclude that when we have relatively sufficient labeled data, DASC on 'small' LMs can be a competitive option compared with LLMs like ChatGPT that can not be easily tuned.
>
> ---
>
> *Experimental Details*: Specifically, we encode the attribute values into the ChatGPT system message with the template following the popular role play format (ref. https://github.com/f/awesome-chatgpt-prompts):
> ```
> f"A dialog history is given below. Please act as the {current speaker} and respond to the next sentence with a {dialog act} in the voice of {gender and emotion}."
> ```
> Then we use dialog history as the user message, and let ChatGPT give a following sentence. We use temperature=0 and stop="\n", that is, greedy search for one line of text, similar to the setting of the original dataset.
>
> ### 2. The classifier of determining attributes using only parts of the sentence may not perform well. Specifically, I am wondering what is the performance of the attribute classifer obtained using Eq.2 and Eq.7.
>
> This is a good question. We didn't deliberately evaluate the accuracy for the attribute classifier as this is not our focus in this work, and such metric will be hard to define and implement. However, we do print out the in-batch prediction AUC (which doesn't require the tuning of threshold) during our experiments at the 1st, 3rd, 11th token. We observed that in most batches, the emotion attribute AUC at the 1st token is about 0.70, while at the 11th token, the AUC generally rises to > 0.9, and in many batches = 1.
>
> We think that it is possible to further improve the model by aligning higher control weights on positions where the model is confident about its attribute prediction. However, we failed to achieve success in our preliminary attempts. We may explore further on this in future works.
>
> ### 3. Some of the experiment results could be explained in more details. For example, the author observes that "Compared to CTRL, DASC has lower Sensibleness but higher Interestingness", but why? Is that because DASC is bad for exhibiting Sensibleness? Similar results are also observed in Table1.
>
> Let's explain this with an example
> ```
> A: Hello, do you have time to chat? I'm a little bit sad.
> B:
> Control Attributes: Gender (female), Emotion (neutral), Question or Statement (statement)
> baseline: I'm ok. What's wrong with you?
> DASC: My mom is a doctor, and my mom doesn't like me very much, and I want her to help her.
> ```
> There are some (but not many) cases like this, where DASC would sacrifice the generation's coherence with the context for a better exploitation of control attributes. In this example, DASC introduced `mom` (a female word), and repeatedly used the word `her` to meet the requirement of generation with the female attribute.

---

### Official Review · Reviewer_kcyB · 2023-08-09

**Soundness:** 4

**Excitement:**

3: Ambivalent: It has merits (e.g., it reports state-of-the-art results, the idea is nice), but there are key weaknesses (e.g., it describes incremental work), and it can significantly benefit from another round of revision. However, I won't object to accepting it if my co-reviewers champion it.

**Paper Topic And Main Contributions:**

This paper proposes a framework for multi-attribute controllable dialogue generation. It uses the weighted decoding paradigm for strong controllability and further grounds it in an attribute semantic space.

**Reasons To Accept:**

1. The paper studies an important multi-attribute controllable dialogue generation task and proposes an intuitive weighted decoding method.
2. The analysis is comprehensive and instructive.

**Reasons To Reject:**

lack of some important baselines, such as FUDGE, PPLM,  Cocon, and recent PLMs.

**Reproducibility:**

5: Could easily reproduce the results.

**Reviewer Confidence:**

3: Pretty sure, but there's a chance I missed something. Although I have a good feel for this area in general, I did not carefully check the paper's details, e.g., the math, experimental design, or novelty.

---

> ### Author Rebuttal · Authors · 2023-08-28
>
> Thanks for your suggestions.
>
> As for the lack of baselines. We didn't experiment with baselines like PPLM because they are initially designed for single-attribute control and would be very costly to extend to multi-attribute cases, requiring multiple classifiers/gradient backward passes. According to previous works, their controllability is also generally worse than DIRECTOR, our major baseline.
>
> We agree that examining the performance of recent PLMs on this task would be interesting. Therefore, we now supplement new experiment results with ChatGPT(gpt-3.5-turbo) on the DuLemon dataset. We will show the results first, and give the experimental details at the bottom.
>
> *Since it is difficult to make fair comparion with prior results using human annotated scores, we just provide the automatic metrics. According to the suggestion of reviewer yeTY, we also added another baseline BoB, which is a representative method with finetuned model that leveraged persona description text for controllable generation.
>
> Results on the test set (ref Table 1).
> |          |   BScore  |   Dist-2  |   Acc_G   |   Acc_E   |    Acc_Q   |
> |:--------:|:---------:|:---------:|:---------:|:---------:|:----------:|
> | Baseline |   68.18   |   19.25   |   68.49   |   46.31   |    69.61   |
> |   CTRL   | **71.09** |   18.91   |   85.32   |   77.49   | **100.00** |
> |   DASC   |   70.42   |   21.94   | **95.85** | **86.07** | **100.00** |
> |   BoB    |   65.47   |   23.44   |   74.59   |   64.76   |    98.51   |
> |  ChatGPT |   66.21   | **30.98** |   69.49   |   56.88   |    98.22   |
>
> Robustness test results (ref Table 3).
> |         |   Dist-2  |   Acc_E   |
> |--------:|:---------:|:---------:|
> |    CTRL |   21.07   |   43.38   |
> |    DASC |   26.71   | **65.38** |
> |     BoB |   24.25   |   30.13   |
> | ChatGPT | **37.02** |   30.00   |
>
> We can see that ChatGPT can generate highly diverse texts. However, ChatGPT cannot achieve comparable controllability with finetuned methods like CTRL and DASC, especially on attributes whose manifestation cannot be easily described in the prompt (e.g. Gender and Emotion).
>
> Therefore, we may conclude that when we have relatively sufficient labeled data, DASC on 'small' LMs can be a competitive option compared with LLMs like ChatGPT which can not be easily tuned.
>
> ---
>
> *Experimental Details*: Specifically, we encode the attribute values into the ChatGPT system message with the template following the popular role play format (ref. https://github.com/f/awesome-chatgpt-prompts):
> ```
> f"A dialog history is given below. Please act as the {current speaker} and respond to the next sentence with a {dialog act} in the voice of {gender and emotion}."
> ```
> Then we use dialog history as the user message, and let ChatGPT give a following sentence. We use temperature=0 and stop="\n", that is, greedy search for one line of text, similar to the setting of the original dataset.

---

### Meta-Review · Area_Chair_UNF3 · 2023-09-19

**Recommendation:** 3

**Metareview:**

The authors propose a new LLM decoding method that allows to control for multiple attributes. The reviewers raise concerns on comparisons with related works and ablation studies, to which the authors respond by conducting additional experiments, showing that their method outperforms the new baselines on most metrics. On the other hand, the reviewers appreciate the novelty and performance of the proposed method and for all the reasons above I would recommend accepting this paper.

---

### Decision · Program_Chairs · 2023-10-07

**Decision:**

Accept-Main

**Comment:**

The authors propose a new LLM decoding method that allows to control for multiple attributes. The reviewers raise concerns on comparisons with related works and ablation studies, to which the authors respond by conducting additional experiments, showing that their method outperforms the new baselines on most metrics. On the other hand, the reviewers appreciate the novelty and performance of the proposed method and for all the reasons above I would recommend accepting this paper.